# Characterization of Worldwide Olive Germplasm Banks of Marrakech (Morocco) and Córdoba (Spain): Towards management and use of olive germplasm in breeding programs

Ahmed El Bakkali[1]*, Laila Essalouh[2,3], Christine Tollon[2], Ronan Rivallan[2], Pierre Mournet[2], Abdelmajid Moukhli[4], Hayat Zaher[4], Abderrahmane Mekkaoui[1], Amal Hadidou[1], Lhassane Sikaoui[4], Bouchaib Khadari[2,5]*

**1** INRA, UR Amélioration des Plantes et Conservation des Ressources Phyto-génétiques, Meknès, Morocco, **2** AGAP, University Montpellier, CIRAD, INRA, Montpellier SupAgro, Montpellier, France, **3** EPLEFPA de Nîmes-CFPPA du Gard, Rodilhan, France, **4** INRA, UR Amélioration des Plantes, Marrakech, Morocco, **5** Conservatoire Botanique National Méditerranéen de Porquerolles (CBNMed), UMR AGAP, Montpellier, France

* khadari@supagro.fr (BK); ahmed_elbakkali@yahoo.fr (AEB)

## Abstract

Olive (*Olea europaea* L.) is a major fruit crop in the Mediterranean Basin. *Ex-situ* olive management is essential to ensure optimal use of genetic resources in breeding programs. The Worldwide Olive Germplasm Bank of Córdoba (WOGBC), Spain, and Marrakech (WOGBM), Morocco, are currently the largest existing olive germplasm collections. Characterization, identification, comparison and authentication of all accessions in both collections could thus provide useful information for managing olive germplasm for its preservation, exchange within the scientific community and use in breeding programs. Here we applied 20 microsatellite markers (SSR) and 11 endocarp morphological traits to discriminate and authenticate 1091 olive accessions belonging to WOGBM and WOGBC (554 and 537, respectively). Of all the analyzed accessions, 672 distinct SSR profiles considered as unique genotypes were identified, but only 130 were present in both collections. Combining SSR markers and endocarp traits led to the identification of 535 cultivars (126 in common) and 120 authenticated cultivars. No significant differences were observed between collections regarding the allelic richness and diversity index. We concluded that the genetic diversity level was stable despite marked contrasts in varietal composition between collections, which could be explained by their different collection establishment conditions. This highlights the extent of cultivar variability within WOGBs. Moreover, we detected 192 mislabeling errors, 72 of which were found in WOGBM. A total of 228 genotypes as molecular variants of 74 cultivars, 79 synonyms and 39 homonyms as new cases were identified. Both collections were combined to define the nested core collections of 55, 121 and 150 sample sizes proposed for further studies. This study was a preliminary step towards managing and mining the genetic diversity in both collections while developing collaborations between olive

**Data Availability Statement:** All relevant data are within the paper and its Supporting Information files.

**Funding:** This work was supported by BeFOre project N. 645595 (Bioresources for Oliviculture)/ H2020-MSCA-RISE-2014 Marie Skłodowska-Curie Research and Innovation Staff Exchange Agropolis Fondation OliveMed N. 1202-066 through the "Investissements d'avenir"/Labex Agro ANR-10-Labex-0001-01 program. The funders had no role in the study design, data collection and analysis, decision to publish, or preparation of the manuscript.

**Competing interests:** The authors have declared that no competing interests exist.

research teams to conduct association mapping studies by exchanging and phenotyping accessions in contrasted environmental sites.

## Introduction

Olive (*Olea europaea*, ssp. *europaea*, var. *europaea* [1]) is widely cultivated for oil and canned fruit. It represents a commercially important fruit crop in the Mediterranean Basin, where about 95% of the world's olives are produced. More than 3,300,000 t of olive oil are produced annually throughout the world [2] on an area of over 10.8 million ha [3], ranking 7th among all vegetable oils produced worldwide, while olive ranks 25th among the 160 most cultivated crops in the world [3]. Of the 47 olive growing countries, Spain, Italy and Greece are the top three countries, accounting for around 38%, 13% and 10% of total olive oil production, respectively [2]. Cultivated olive was domesticated from the wild type (*Olea europaea* subsp. *Europaea* var. *sylvestris*) in the Middle East 6,000 years ago [4,5]. Early domesticated forms were probably disseminated during successive human migrations from east to west and introgressed with local wild olives, in turn giving rise to local cultivated forms through selection by farmers [6–10]. Olive spread from Mediterranean areas throughout the world. This crop is now of increasing commercial interest beyond the Mediterranean Basin, including countries such as Australia, Chile and USA [2].

Over the long history of olive domestication, cultivated forms have been selected, propagated and disseminated by farmers. An international initiative was thus conducted to pool olive germplasm information in a single database. The 2008 web-based edition (http://www.oleadb.it/) is currently the largest database with information extracted from almost 1520 publications [11]. This database nevertheless represents an underestimated level of domesticated olive diversity since it overlooks many minor local cultivars specific to olive growing areas such as Morocco, Cyprus and Syria. There are currently more than 1,200 cultivars with over 3,000 synonyms reported in 54 different countries and maintained in almost 100 separate collections at international, national and regional levels for conservation and evaluation purposes [11–13], this includes: in the Mediterranean Basin, at Cosenza (Italy, 500 cultivars [14]), Izmir (Turkey, 96 cultivars [15]) and Chania (Greece, 47 cultivars [16]), in addition to new olive-growing regions of the world such as at Davis (USA, [17]) and Mendoza (Argentina, [18]). However, few of these cultivars have been fully characterized using molecular markers and morphological traits. Two worldwide olive germplasm banks (WOGB) exist in the Mediterranean Basin, the first was set up in the 1970s at Cordoba (Spain, WOGBC), with about 500 accessions from 21 countries [19–21]. In 2003, in the framework of the ResGen-T96/97 Project funded by the EU and IOOC and including 16 partners, a second WOGB was created at the INRA Research Station of Tassaoute, Marrakech (Morocco, WOGBM). This worldwide collection presently includes almost 560 accessions originating from 14 Mediterranean countries [22–23]. To optimize olive germplasm sampling, local genetic resources had been characterized by different partners using standardized morphological and/or molecular descriptors. WOGBM was thus established while including previously characterized olive genetic resources from each Mediterranean country.

Recently, socioeconomic changes in most olive producing countries have led to significant improvements in olive growing, including the establishment of modern orchards based exclusively on a few high-yielding and low-vigor cultivars, such as cv. Arbequina. These trends may potentially lead to the erosion of local olive germplasm because several minor traditional

cultivars are being replaced by a few cultivars. In the current setting of climate change [24] and the emergence of new diseases, such as *Xylella fastidiosa* [25], olive cropping systems based solely on a few cultivars may be less resilient than conventional systems. Therefore, the preservation of genetic diversity in *ex-situ* collections could provide material to substantially enhance modern orchards and breeding programs. The identification and authentication of accessions in collections is therefore considered essential prior to any use of olive germplasm.

As a clonally propagated fruit crop, olive trees have been widely disseminated throughout the Mediterranean Basin and in new olive-growing regions worldwide, leading in many cases to synonymy (different names used for the same cultivar) [26], homonymy (the same name used for different cultivars) [27] and molecular variants, i.e. intra-cultivar variation [28–30]. Moreover, mislabeling errors within collections, which may occur at any step during the plant material processing, is another serious problem in germplasm collection management [21]. Such constraints have also been identified in other clonally propagated fruit crops such as grape [31], fig [32] and apple [33]. This issue could be overcome via cultivar characterization and identification, thus avoiding varietal confusion and providing authenticated cultivars.

Olive cultivar identification was first based solely on morphological and agronomical traits. Descriptors were then used for the characterization of olive flowers, pollen, leaves, fruit and endocarp, oil content, vigor, yield and flowering phenology [34–40]. Although these descriptors are very useful for field surveys and germplasm bank characterization [18,21,41,42], they were found to be only partially informative due to their limited number and the fact that they were impacted by environment conditions. Molecular markers were then applied to better characterize and clearly identify olive cultivars. Molecular studies were initially conducted using isoenzyme markers [43,44] and then DNA-based markers, including restriction fragment length polymorphism (RFLP) [45], random amplified polymorphic DNA (RAPD) [46,47], inter-simple sequence repeat (ISSR) [48–49], amplified fragment length polymorphism (AFLP) [50], simple sequence repeat (SSR) [17,21,22,30,51–55], and single nucleotide polymorphism (SNP) [19,56–58]. Recent available information on olive genome assembly [59,60] will broaden the prospects for varietal identification using DNA-based fingerprinting with EST-SSR and SNP markers [61–63].

Most olive cultivar characterization and identification studies have been carried out independently [17,21–23,51,64]. Although the same panels of SSR loci have been used, attempts to align databases of different collections have led to discrepancies among accessions, which could be mainly explained by the nature of the SSR loci studied as most of them have di-nucleotide repeats (AG/CT) [65]. As reported by Weeks et al. [66], 83% of between-laboratory discrepancies in scoring di-nucleotide microsatellites have been due to erroneous length attributions during the binning process.

With the aim of generating a complete database of identified and authenticated cultivars from the Mediterranean Basin by analyzing the world's largest collections, here for both WOGBC and WOGBM, we applied an approach for the characterization, identification and authentication similar to that previously described by Trujillo et al. [21] on WOGBC. Using 33 selected SSR markers and 11 endocarp traits on 824 olive trees, representing 499 accessions from 21 countries, the latter authors identified 332 cultivars in which 200 were considered as authenticated cultivars, and a total of 537 profiles (based on SSR loci and endocarp descriptors) were revealed.

Here we sought answers to two main questions: (i) could pooling the two WOGBs increase the level of variability contained in the conserved genetic resources? and (ii) would pooling of the two WOGB collections have an impact on defining a core collection suitable for association mapping? We thus applied 20 SSR loci and 11 endocarp traits in order: (i) to characterize, identify and authenticate olive cultivars within both WOGBs, (ii) to establish one consensus

olive dataset for both WOGBs using SSR markers and morphological traits to manage, use and exchange plant material, (iii) to propose a subset of cultivars encompassing all of the genetic diversity in both collections, and (iv) to release information and methodologies that could be used for characterizing other national and international *ex-situ* olive collections [14–17,47,49,67].

## Material & methods

### Plant material

The study was carried out using 554 accessions from WOGBM, identified by single codes, corresponding to 486 denominations from 14 countries (Table 1 and S1 Table). For WOGBC, a total of 537 olive trees with different profiles, as identified by Trujillo et al. [21], corresponding to 499 accessions and 405 denominations from 21 countries, was used (Table 1 and S1 Table). These 537 profiles were identified based on a total of 824 olive trees using 33 SSR markers and 11 endocarp traits [21].

**Table 1. Number of olive accessions compared per country in both collections and the number of genotypes including variants and cultivars that were identified and authenticated.** WOGBM (M), WOGBC (C), Total (T), shared between both collections (Both).

| | Origin | No. of trees | | | No. of accessions | | | | No. of genotypes[1] | | | | No. of identified cultivars[2] | | | | No. of authentic cultivars[3] | | | |
|---|---|---|---|---|---|---|---|---|---|---|---|---|---|---|---|---|---|---|---|---|
| | | M | C | T | M | C | T | Both[a]()[b] | M | C | T | Both | M | C | T | Both | M | C | T | Both |
| 1 | Albania | | 13 | 13 | | 12 | 12 | | 1 | 11 | 11 | 1 | | 10 | 10 | | | 3 | 3 | |
| 2 | Argentina | | 2 | 2 | | 2 | 2 | | | 1 | 1 | | | 1 | 1 | | | | | |
| 3 | Algeria | 43 | 3 | 46 | 43 | 2 | 45 | 2(4) | 27 | 1 | 27 | 1 | 26 | 1 | 26 | 1 | 1 | 1 | 1 | 1 |
| 4 | Chile | | 1 | 1 | | 1 | 1 | | | 1 | 1 | | | 1 | 1 | | | 1 | 1 | |
| 5 | Cyprus | 31 | 3 | 34 | 31 | 3 | 34 | 3(9) | 4 | 2 | 6 | | 1 | 1 | 1 | 1 | 1 | 1 | 1 | 1 |
| 6 | Croatia (HRV) | 16 | 7 | 23 | 16 | 7 | 23 | 4(9) | 10 | 7 | 14 | 4 | 9 | 7 | 13 | 3 | 3 | 5 | 5 | 3 |
| 7 | Egypt | 19 | 5 | 24 | 19 | 5 | 24 | 4(8) | 17 | 3 | 20 | | 17 | 3 | 20 | | | | 1 | |
| 8 | France | 13 | 13 | 26 | 13 | 10 | 23 | 8(18) | 9 | 10 | 13 | 6 | 8 | 8 | 11 | 5 | 5 | 6 | 6 | 5 |
| 9 | Greece | 17 | 20 | 37 | 17 | 18 | 35 | 7(15) | 14 | 17 | 26 | 5 | 13 | 15 | 22 | 6 | 6 | 11 | 11 | 6 |
| 10 | Iran | | 5 | 5 | | 5 | 5 | | | 5 | 5 | | | 5 | 5 | | | 5 | 5 | |
| 11 | Israel | | 9 | 9 | | 9 | 9 | | | 3 | 3 | | | 3 | 3 | | | 2 | 2 | |
| 12 | Italy | 163 | 40 | 203 | 163 | 36 | 199 | 16(47) | 128 | 30 | 146 | 12 | 92 | 20 | 100 | 12 | 12 | 17 | 17 | 12 |
| 13 | Lebanon | 16 | 2 | 18 | 16 | 2 | 18 | 2(13) | 11 | 2 | 13 | 3 | 4 | 1 | 4 | 1 | 1 | 1 | 1 | 1 |
| 14 | Mexico | | 7 | 7 | | 7 | 7 | | | 2 | 2 | | | 2 | 2 | | | | | |
| 15 | Morocco | 27 | 4 | 31 | 27 | 4 | 31 | 3(8) | 11 | 3 | 12 | 2 | 10 | 1 | 10 | 1 | 1 | 1 | 1 | 1 |
| 16 | Portugal | 15 | 11 | 26 | 15 | 10 | 25 | 7(14) | 10 | 8 | 14 | 5 | 10 | 6 | 12 | 4 | 4 | 4 | 4 | 4 |
| 17 | Slovenia | 10 | | 10 | 10 | | 10 | 1(3) | 3 | 1 | 3 | 1 | 3 | | 3 | | | | | |
| 18 | Spain | 89 | 298 | 387 | 89 | 279 | 368 | 91(220) | 100 | 232 | 247 | 85 | 86 | 186 | 191 | 81 | 81 | 136 | 136 | 81 |
| 19 | Syria | 70 | 61 | 131 | 70 | 56 | 126 | 24(67) | 42 | 42 | 80 | 5 | 35 | 37 | 64 | 8 | 2 | 2 | 2 | 2 |
| 20 | Tunisia | 25 | 7 | 32 | 25 | 7 | 32 | 6(13) | 16 | 6 | 19 | 3 | 14 | 6 | 18 | 2 | 2 | 5 | 6 | 2 |
| 21 | Turkey | | 20 | 20 | | 19 | 19 | | 2 | 17 | 17 | 2 | 1 | 15 | 15 | 1 | 1 | 8 | 8 | 1 |
| 22 | USA | | 4 | 4 | | 4 | 4 | | | 2 | 2 | | | 2 | 2 | | | | | |
| 23 | Unkown | | 2 | 2 | | 1 | 1 | | | 1 | 1 | | | 1 | 1 | | | | | |
| | **Total** | **554** | **537** | **1091** | **554** | **499** | **1053** | **178(448)** | **402** | **400** | **672** | **130** | **329** | **332** | **535** | **126** | **120** | **210** | **211** | **120** |

[1] based on 20 SSR loci.

[2] based on both SSR loci and endocarp traits.

[3] based on comparison with WOGBC.

[a] Number of similar denominations in both collections.

[b] Number of accessions with similar names in both collections.

## Characterization of WOGBM and WOGBC

**SSR genotyping of WOGBM.** For the 554 WOGBM accessions, total DNA was extracted from 1 g young leaves, as described in Khadari et al. [30]. DNA quality was checked on 0.8% agarose gel, whereas the DNA concentration was estimated using spectrofluorometry (GENios Plus, TECAN, Grödig, Austria).

Twenty SSR loci were used in this study: DCA01, DCA03, DCA04, DCA05, DCA08, DCA09, DCA10, DCA11, DCA15, DCA16 and DCA18 [68]; GAPU59, GAPU71A, GAPU71B, GAPU101 and GAPU103A [69]; UDO99-11, UDO99-17 and UDO99-43 [28] and EMO90 [70]. These markers were selected based on their clear amplification, high polymorphism and reproducibility, as observed by Trujillo et al. [21] and El Bakkali et al. [23]. PCR amplification was carried out in a total volume of 20 μl containing 20 ng of genomic DNA, 1x PCR buffer, 1.5 mM MgCl$_2$, 0.2 M of each dNTP, 0.1 U of *Taq* DNA polymerase, and 2 pmol of forward (fluorescent labelled) and reverse primers. PCR reactions were performed in a thermal cycler (Mastercycler ep gradient S) at 94°C for 5 min, followed by 35 denaturation cycles at 94°C for 30 s, 50, 55 or 57°C for 1 min for annealing, depending on the locus, and 72°C extension for 1 min. A post-thermocycling 10 min extension at 72°C was carried out. PCR products were separated using an automatic capillary sequencer (ABI prism 3130XL Genetic Analyzer Applied Biosystems, Foster City, CA, USA), using GeneScan 400 HD-Rox as internal standard, and chromatograms were then visualized and analyzed with GeneMapper 3.7 software (Applied Biosystems).

**Aligning SSR alleles of WOGBM and WOGBC databases.** To align alleles between the two collections, 47 accessions from WOGBC already analyzed by Trujillo et al. [21] were re-genotyped with the 20 SSR loci in laboratory conditions similar to those used for genotyping and visualizing WOGBM accessions (S2 Table). A total of 407 alleles among a set of 466 alleles (87.3%) were observed within this panel of 47 accessions using 33 SSR markers from the previous study of Trujillo et al. [21]. Once the 47 accessions were genotyped, the sizes of alleles observed in WOGBM were adjusted to match those recorded by Trujillo et al. [21] in WOGBC (S2 Table).

**Morphological characterization.** Morphological characterization was independently carried out by two observers using a representative sample of 40 endocarps/tree for 518 olive trees among the 554 analyzed with SSR markers. Each tree was analyzed twice during 2015 and 2016. Based on the protocol described by Trujillo et al. [21], we used eleven endocarp traits to characterize WOGBM and to compare the morphological datasets of both collections: weight, shape in position A, symmetry in positions A and B, position of maximum transverse diameter in position B, shape of apex in position A, shape of base in position A, surface roughness, number of grooves on the basal end, distribution of the grooves on the basal end and presence of mucro.

## Data analysis

Total genotypes detected within the whole dataset (1091 olive trees) were discriminated by pairwise comparison of their SSR profiles using the Excel Microsatellite Toolkit [71]. Genetic diversity in each collection separately and in the whole dataset (both collections) was estimated by calculating different parameters for each microsatellite locus using the Excel Microsatellite Toolkit, including: allele size (bp), number of alleles (Na), number of alleles observed once (Nu), observed (Ho) and expected heterozygosity (He; [72]) and polymorphism information content (PIC; [73]). Otherwise, pairwise comparisons between samples based on endocarp traits were conducted to identify similar morphological profiles using a binary matrix of different morphological states.

Genetic structure in the whole dataset and for each collection was determined using a model-based clustering method implemented in STRUCTURE v.2.3.4 [74]. Bayesian analysis was run under the admixture model for a burn-in period of 200,000 iterations and a post-burning simulation length of 1,000,000 while assuming a correlation among allele frequencies. Analyses were run for $K$ clusters from 1 to 8 with 10 replicates per $K$ value. The most likely number of clusters was determined using the ad-hoc $\Delta K$ measure [75] with the R program [76], whereas the similarity index between the 10 replicates for the same $K$ clusters ($H'$) was calculated with the CLUMPP v1.1.2 program (Greedy algorithm [77]).

To describe the spatial distribution of genotypes, a principal coordinate analysis (PCoA), as implemented in the DARWIN v.5.0.137 program [78], was constructed based on SSR data with the simple matching coefficient [79]. For genotypes showing molecular variants in two collections, SSR data were converted into binary matrix (0 and 1) and the dendrogram was generated using the Dice similarity index [80] and UPGMA method with NTSYs-PC v2.02 software [81].

Comparisons between the two collections were carried out based on: (1) the accession denomination, (2) number of genotypes and cultivars in common, (3) genetic parameters such as number of alleles (Na) and Nei diversity index (He), (4) the genetic distance distribution between shared genotypes and those specific to each collection using the Smouse and Peakall index [82] in GENALEX 6.5 [83], (5) the allelic richness (Ar [84]); and (6) the genetic structure within both collections using the model-based Bayesian clustering approach implemented in STRUCTURE. The allelic richness (Ar) was computed according to a generalized rarefaction approach at the standardized G value using the ADZE program [85]. Significant differences in rarefied Ar and He were determined using the Mann-Whitney comparison test ($p \leq 0.05$) with the PAST program [86].

Core collections representative of the genetic diversity in the whole dataset were constructed based on the two-step method, as described by El Bakkali et al. [23], by combining approaches implemented in MSTRAT [87] and CORE HUNTER [88] programs, and using Maximization [89] and '*Sh*' strategies, respectively. Fifty final core collections were generated independently and one core collection was arbitrary selected and described.

## Results

### Characterization of WOGBM

**SSR polymorphism.** Using the 20 SSR loci, a total of 370 alleles were observed within the WOGBM collection among which 49 alleles were just observed once. These alleles were checked by re-amplification to determine their correct SSR profiles. The number of alleles ranged from 6 for DCA15 to 35 for DCA10, with a mean of 18.5 alleles per locus (Table 2). Allele frequencies ranged from 0.09% to 70.5%, while 203 alleles (54.8%) showed a frequency of less than 1%.

The observed heterozygosity (Ho) ranged from 0.309 (DCA10) to 0.973 (UDO-11), with a mean of 0.758. He ranged from 0.473 (GAPU-71A) to 0.878 (DCA09) with a mean of 0.762. Eighteen markers among the 20 had a PIC of above 0.5 (Table 2).

**Cultivar identification using SSR markers and morphological traits.** The 20 SSR markers identified 402 unique genotypes among the 554 WOGM accessions with at least one dissimilar allele (Table 1). In line with the findings of Trujillo et al. [21], each genotype was coded with an ordinal number (S1 and S3 Tables). The 554 accessions were classified as follows: (i) 323 accessions were identified as unique SSR profiles (not duplicated in WOGBM), and (ii) 231 accessions had SSR profiles in common with other accessions in the collection, resulting in the identification of 79 different SSR profiles.

**Table 2. Summary of genetic diversity parameters of 20 SSR markers observed in the WOGBM and WOGBC collections.** Number of alleles (Na), number of alleles observed once (Nu), allelic richness (Ar), expected (He) and observed (Ho) heterozygosity, and polymorphic information content (PIC).

| | WOGBM | | | | | | | WOGBC | | | | | | | Whole dataset | | | | | | | |
|---|---|---|---|---|---|---|---|---|---|---|---|---|---|---|---|---|---|---|---|---|---|---|
| | Size (bp) | Na | Nu | Ar[1] | Ho | He | PIC | Size (bp) | Na | Nu | Ar[1] | Ho | He | PIC | Size (bp) | Na | Na[2] | Nu | Nu[2] | Ho | He | PIC |
| DCA01 | 204–274 | 21 | 7 | 15.7 | 0.734 | 0.622 | 0.574 | 204–274 | 14 | 3 | 11.1 | 0.787 | 0.624 | 0.565 | 204–274 | 21 | 14 | 3 | 6 | 0.760 | 0.624 | 0.571 |
| DCA03 | 227–255 | 13 | 2 | 11.6 | 0.910 | 0.854 | 0.836 | 227–255 | 15 | 1 | 13.7 | 0.937 | 0.843 | 0.823 | 227–255 | 15 | 13 | | 2 | 0.923 | 0.850 | 0.832 |
| DCA04 | 116–198 | 32 | 2 | 28.6 | 0.621 | 0.848 | 0.831 | 116–198 | 28 | 1 | 24.5 | 0.630 | 0.801 | 0.782 | 116–198 | 35 | 25 | 3 | | 0.625 | 0.829 | 0.811 |
| DCA05 | 191–213 | 12 | | 11.3 | 0.498 | 0.489 | 0.473 | 191–211 | 10 | | 9.9 | 0.378 | 0.387 | 0.376 | 191–213 | 12 | 10 | | | 0.439 | 0.440 | 0.428 |
| DCA08 | 123–163 | 21 | 3 | 18.3 | 0.764 | 0.840 | 0.819 | 123–168 | 18 | 2 | 16.0 | 0.597 | 0.801 | 0.773 | 123–168 | 23 | 16 | 3 | 1 | 0.684 | 0.826 | 0.804 |
| DCA09 | 160–218 | 25 | 2 | 23.1 | 0.946 | 0.878 | 0.865 | 160–214 | 22 | 2 | 20.4 | 0.955 | 0.863 | 0.847 | 160–218 | 26 | 21 | 2 | 1 | 0.951 | 0.877 | 0.860 |
| DCA10 | 138–263 | 35 | 3 | 31.0 | 0.309 | 0.857 | 0.845 | 138–260 | 36 | 3 | 31.4 | 0.229 | 0.800 | 0.783 | 138–263 | 41 | 30 | 3 | 3 | 0.271 | 0.833 | 0.820 |
| DCA11 | 126–182 | 23 | | 21.0 | 0.914 | 0.827 | 0.804 | 126–185 | 26 | 4 | 22.1 | 0.922 | 0.810 | 0.783 | 126–185 | 26 | 23 | 2 | 2 | 0.918 | 0.819 | 0.795 |
| DCA15 | 243–267 | 6 | | 5.5 | 0.432 | 0.569 | 0.518 | 243–267 | 7 | 1 | 6.2 | 0.261 | 0.524 | 0.469 | 243–267 | 7 | 6 | 1 | | 0.348 | 0.550 | 0.499 |
| DCA16 | 122–230 | 34 | 9 | 26.9 | 0.960 | 0.867 | 0.852 | 122–228 | 32 | 10 | 24.3 | 0.965 | 0.850 | 0.832 | 122–230 | 39 | 27 | 6 | 10 | 0.962 | 0.861 | 0.845 |
| DCA18 | 154–189 | 17 | 2 | 15.6 | 0.865 | 0.817 | 0.797 | 158–193 | 16 | 1 | 15.0 | 0.927 | 0.820 | 0.796 | 154–193 | 19 | 14 | 3 | | 0.896 | 0.823 | 0.801 |
| EMO90 | 181–208 | 9 | | 8.6 | 0.739 | 0.675 | 0.639 | 181–208 | 10 | 2 | 9.0 | 0.684 | 0.637 | 0.600 | 181–208 | 10 | 9 | 1 | 1 | 0.712 | 0.659 | 0.624 |
| GAPU59 | 206–239 | 11 | 3 | 9.3 | 0.626 | 0.621 | 0.579 | 194–226 | 10 | 2 | 8.8 | 0.637 | 0.608 | 0.565 | 194–239 | 13 | 8 | 4 | 1 | 0.631 | 0.616 | 0.574 |
| GAPU71A | 206–256 | 16 | 6 | 11.4 | 0.571 | 0.473 | 0.422 | 206–246 | 10 | 3 | 7.8 | 0.471 | 0.430 | 0.372 | 206–256 | 16 | 10 | 3 | 4 | 0.522 | 0.452 | 0.398 |
| GAPU 71B | 118–147 | 9 | | 8.4 | 0.899 | 0.803 | 0.772 | 118–147 | 9 | | 8.4 | 0.940 | 0.801 | 0.770 | 118–147 | 10 | 8 | | | 0.919 | 0.803 | 0.773 |
| GAPU101 | 183–219 | 13 | 1 | 11.8 | 0.948 | 0.852 | 0.834 | 183–219 | 13 | 1 | 12.0 | 0.979 | 0.833 | 0.810 | 183–219 | 14 | 12 | 1 | 1 | 0.963 | 0.845 | 0.826 |
| GAPU103A | 133–194 | 26 | 5 | 21.3 | 0.820 | 0.849 | 0.832 | 133–208 | 26 | 4 | 22.3 | 0.734 | 0.814 | 0.789 | 133–208 | 30 | 22 | 5 | 3 | 0.777 | 0.836 | 0.817 |
| UDO99-11 | 103–140 | 14 | 1 | 12.6 | 0.973 | 0.849 | 0.831 | 103–142 | 14 | 3 | 12.0 | 0.952 | 0.830 | 0.809 | 103–142 | 16 | 12 | 2 | 1 | 0.962 | 0.842 | 0.823 |
| UDO99-17 | 152–173 | 7 | | 7.0 | 0.775 | 0.782 | 0.749 | 152–173 | 6 | | 6.0 | 0.775 | 0.785 | 0.751 | 152–173 | 7 | 6 | | | 0.775 | 0.784 | 0.751 |
| UDO99-43 | 166–225 | 26 | 3 | 23.0 | 0.870 | 0.876 | 0.863 | 162–225 | 24 | 1 | 21.9 | 0.898 | 0.868 | 0.854 | 162–225 | 27 | 23 | 1 | 2 | 0.883 | 0.875 | 0.862 |
| Mean | | 18.50 | 2.45 | 16.1[a] | 0.758 | 0.762[b] | 0.737 | | 17.30 | 2.20 | 15.1[a] | 0.732 | 0.736[b] | 0.707 | | 20.35 | 15.45 | 2.15 | | 0.746 | 0.752 | 0.725 |
| Total | | 370 | 49 | 321 | | | | | 346 | 44 | 302 | | | | | 407 | 309 | 43 | 38 | | | |

[1]Computed at G value of 400. No significant difference between both collections (Mann-Whitney test, p-value>0.05).

[2]Shared alleles between both collections.

[a,b]Index of significance at p-value < 0.05.

Three SSR markers (DCA09, UDO099-043 and DCA16) were able to identify 84% of the accessions. Six markers (DCA09, UDO099-043, DCA16, DCA10, DCA04 and GAPU103) were used to discriminate 95% of the accessions, whereas eight additional markers (DCA03, DCA08, DCA18, GAPU71B, DCA11, UDO099-011, DCA01 and DCA15) were applied to distinguish all WOGBM accessions.

Based on endocarp traits, a total of 251 different morphological profiles were identified that were coded with an ordinal number (S1 and S4 Tables). The accessions were classified as follows: (i) 164 accessions had unique morphological profiles (not duplicated in WOGBM), and (ii) 354 accessions had morphological profiles in common with other accessions in the collection, resulting in the identification of 87 different morphological profiles.

Then 329 different olive cultivars were identified using 20 SSR markers and 11 endocarp traits (Table 1; S1 and S4 Tables). SSR markers were used as the main criteria for cultivar identification for the 36 accessions with no endocarp trait data (S1 Table). Among all genotype pairs in WOGBM, only 5 were observed which had identical SSR profiles but distinct morphological profiles: "Fouji vert" (MAR00291)/"Besbessi" (MAR00464), "Lechin de Sevilla" (MAR00243)/"Zarza" (MAR00280), "Beladi" (MAR00572)/"Beladi Aitaroun" (MAR00577), "Olivastra di Montalcino" (MAR00375)/"Mortellino" (MAR00373) and "Beldi" (MAR00288)/ "Giarfara" (MAR00106; S1 Table).

## Characterization of WOGBC

**SSR polymorphism.** A total of 346 alleles were observed in WOGBC, with a mean of 17.3 alleles per locus, while 189 of these alleles (54.62%) showed a frequency of less than 1%, and 44 alleles were just observed once (Table 2). The allele number ranged from 6 for UDO99-17 to 36 for DCA10 loci.

Ho ranged from 0.229 (DCA10) to 0.979 (GAPU101), with a mean of 0.732. He ranged from 0.387 (DCA05) to 0.868 (UDO099-43), with a mean of 0.736. Seventeen among the 20 markers had a PIC of above 0.5 (Table 2).

**Cultivar identification using SSR markers and morphological traits.** Using 20 SSR markers, the 537 WOGBC profiles were classified in 400 different genotypes coded with an ordinal number, as reported by Trujillo et al. [21] (S1 and S3 Tables). Only 15 SSR profiles were switched using the current set of markers compared to 33 analyzed by Trujillo et al. [21] (S1 Table). For instance, Trujillo et al. [21] identified 239 different genotypes originating from Spain whereas the 20 SSR markers only revealed 232 genotypes in the current analysis. Furthermore, the "Alameño de Marchena" (COR000254) and "Zarza" (COR000038) accessions were identified by Trujillo et al. [21] as being different from "Picholine Marocaine" and "Lechín de Sevilla", respectively. They had two dissimilar alleles at GAPU82 and UDO-42 loci that were not taken into account in the current set of markers for the first accession, and at UDO-05 for the second. Otherwise, among the 400 different genotypes, 320 were identified as unique SSR profiles (not duplicated) whereas 217 accessions had SSR profiles in common, resulting in the identification of 80 different SSR profiles.

Based only on endocarp traits, a total of 245 different morphological profiles were identified in WOGBC (S1 and S4 Tables). No morphological data were available for 31 accessions. The 506 olive trees were classified as follows: (i) 147 olive trees had unique morphological profiles (not duplicated in WOGBC), and (ii) 359 had morphological profiles in common with other accessions in the collection, resulting in the identification of 98 different morphological profiles. Each of the following eight pairs of accessions showed similar or nearly identical SSR profiles but presented morphological differences: "Zarza" (COR000038)/"Lechin de Sevilla" (COR000005), "Menya" (COR000669)/"Manzanilla Picua" (COR000377)/"Menya de Reus"

(COR001071), "Chemlali" (COR000744)/"Chetoui" (COR000113), "Gatuno" (COR000380)/
"Abbadi Abou Gabra" (COR001033), "Pulazeqin" (COR001085)/"Itrana" (COR000068),
"Azulejo" (COR000959)/"Negrinha" (COR000123), "Dulzal de Carmona" (COR000031)/
"Imperial de Jaén" (COR000030) and "Jabaluna" (COR000392)/"Escarabajuelo de Úbeda"
(COR000353). By combining both SSR and morphological data, a total of 332 cultivars were
identified in WOGBC (Table 1).

## Comparisons between the two WOGBs

**Based on accession denominations.**   When focusing on the accession denominations, 713
accession names in both collections were listed, corresponding to 1037 accessions, i.e. 495 in
WOGBC and 542 in WOGBM, while 16 accessions had no denominations and were assigned
codes such as "Unkown-OMDZ" (MAR00536) and "Klon-14" (COR001081). Among the 713
denominations, 178 that were found to be common in both collections corresponded to a total
of 448 accessions, i.e. 207 observed only in WOGBM and 241 in WOGBC (Table 1). A total of
308 denominations (335 accessions) and 227 (254 accessions) were specific to WOGBM and
WOGBC, respectively. The highest number of shared denominations was observed in Spanish
germplasm (91; 51%). The other accessions were classified as follows: 1 from Slovenia, 2 (Alge-
ria and Lebanon), 3 (Cyprus and Morocco), 4 (Egypt and Croatia), 6 (Tunisia), 7 (Greece and
Portugal), 8 (France), 16 (Italy) and 24 (Syria; Table 1).

**Based on SSR markers and morphological traits.**   In the combined WOGBM and
WOGBC dataset, 407 alleles were revealed using 20 SSR markers, with a mean of 20.35 alleles
per locus. Among the 407 alleles, 309 alleles were in common, whereas 61 and 37 alleles were
specific to WOGBM and WOGBC, respectively (Table 2). A total of 43 unique alleles
(observed once) were detected in 54 genotypes in both collections (27 in WOGBM, 21 in
WOGBC and 6 in common). No significant difference was observed between the two collec-
tions regarding the allelic richness, computed at a G value of 400, and the diversity index (He;
Mann-Whitney test p-value > 0.05; Table 2).

Similar pairwise genotype patterns were observed in both WOGBs (S1 Fig). In both collec-
tions, only 1054 (0.68%) and 610 (0.42%) pairwise comparisons, for WOGBM and WOGBC,
respectively, represented closely related genotypes that differed by one to four dissimilar
alleles, whereas the remaining pairwise genotypes were distinguished by 5 to 39 dissimilar
alleles. The highest SSR dissimilarity (39 distinct SSR alleles) was observed in only two geno-
type pairs in WOGBM and six pairs in WOGBC.

The analysis of both datasets (1091 olive trees) revealed 672 different SSR profiles. Three
SSR markers (DCA09, DCA16 and UDO099-043) were able to distinguish 77% of the identi-
fied genotypes, whereas six markers (DCA04, DCA09, DCA10, DCA16, UDO099-043 and
GAPU103) could be applied to discriminate 94% of the total identified genotypes. The 672
genotypes were classified as: (i) 130 SSR profiles observed in common between the two collec-
tions, with a total of 436 accessions (213 and 223 for WOGBM and WOGBC, respectively),
and (ii) 542 genotypes were specific to WOGBC (270) or WOGBM (272), corresponding to
655 accessions (341 and 314 for WOGBM and WOGBC, respectively; Tables 1 and 3, S1 and
S3 Tables). The highest number of genotypes in common was identified within Spanish germ-
plasm with 85 genotypes, followed by 12 within Italian germplasm (Table 1).

We identified two genotypes originating from Turkey and one from Albania which were
not represented in WOGBM (Table 1). Genotypes related to the "Gemlik" (COR000092) and
"Samsun Tuzlamalik" (COR000684) accessions from Turkey in WOGBC were actually identi-
fied as being similar to "Safrawi" (MAR00608) and "Ayrouni" (MAR00633)/"Kfar Zita"
(MAR00609), respectively. Similarly, some accessions in WOGBM were found to be close or

**Table 3. Comparison of genotypes shared between the two collections and those specific to each one.** The number of alleles (Na), allelic richness (Ar) and index of diversity (He).

| Group of genotypes | Size of genotypes | N. olive trees | | Na (%)[1] | Ar[2] | He[3] |
|---|---|---|---|---|---|---|
| | | WOGBM | WOGBC | | | |
| **Shared between collections** | 130 | 213 | 223 | 233 (57.2%) | 9.7[a] | 0.724[a] |
| **Specific to WOGBM** | 272 | 341 | | 358 (87.9%) | 12.7[a] | 0.774[b] |
| **Specific to WOGBC** | 270 | | 314 | 329 (80.8%) | 11.5[a] | 0.746[ab] |
| **Mean** | | | | **20.35** | | **0.758** |
| **Total** | **672** | **554** | **537** | **407** | | |

[1] Compared to total number of alleles in both collections (407 alleles).

[2] Computed at G value of 130, no significant difference between both collections (Mann-Whitney comparison test, *p-value* >0.05).

[3] Significant difference between genotypes in common and those specific to WOGBM (Mann-Whitney comparison test, *p-value* <0.05).

[a, b] Index of significance at p-value < 0.05.

similar to some accessions from different countries in WOGBC. For instance, "Dahbia" (MAR00391-Morocco)/"Callosina" (COR000040, COR000060-Spain), "Bouchouk Rkike" (MAR00396-Morocco)/"Ocal" (COR000282-Spain), "Stanboli" (MAR00624-Syria)/"Gordal de Granada" (COR000761-Spain), "Throumbolia" (MAR00184-Greece)/"Grossolana" (MAR0044-Italy)/"Cirujal" (COR000963-Spain) and "Verdial Alentejana" (MAR00213-Portugal)/"Verdial de Huévar" (COR000155-Spain; S1 Table).

In the whole dataset of 672 genotypes, we identified 8 genotypes from different countries which were duplicated. When focusing on these 8 genotypes, the identified discrepancy included: (i) four cases observed only in WOGBC, e.g. "Abbadi Abou Gabra" (COR001033-Syria)/"Gatuno" (COR000380-Spain), "Frantoio A. Corsini" (COR000081-Italy)/"Kokerrmadh Berati" (COR001080-Albania), "Ayrouni" (COR000134-Lebanon)/"Verdial de Huévar" (COR000006-Spain) and "Adramitini" (COR000102-Greece)/"Ayvalik" (COR001474-Turkey); and (ii) four genotypes were shared between the two collections, including "Cordovil de Castelo Branço" (COR000886-Portugal)/"Llorón de Iznalloz" (MAR00342, COR000400-Spain), "Itrana" (MAR0017, COR000068-Italy)/"Pulazeqin" (COR001085-Albania), "Crnica" (COR000734-Croatia)/"Crnica" (MAR00399-Slovenia) and "Analiontas" (MAR00306-Cyprus)/"Sourani Red" (COR001478-Syria; S1 Table).

A total of 233 alleles were observed within the common 130 genotypes, with a mean of 11.65 alleles per locus, whereas 358 and 329 alleles were observed within genotypes specific to WOGBM (272) and WOGBC (270), respectively (Table 3 and S5 Table). A similar genetic distance distribution was observed within the three groups, while being shared and specific to each collection (S2 Fig). The PCoA plot revealed an even distribution of the shared genotypes along both axes accounting for 12.34% of the total genetic variation, whereas those originating from Spain were clustered (S3 Fig). No significant difference in allele number was observed between common genotypes and those specific to each collection regarding allelic richness (Mann-Whitney comparison test; p>0.05). However, a significant difference was observed for He (diversity index) between common genotypes and those specific to WOGBM (p<0.05; Table 3 and S5 Table), while no significant difference was revealed for the He diversity index regarding genotypes specific to each collection.

Based only on endocarp traits, a total of 371 different morphological profiles were identified in both collections (S1 and S4 Tables). Among these accessions: (i) 158 accessions were classified as 120 unique morphological profiles in WOGBC; (ii) 204 accessions as 126 unique

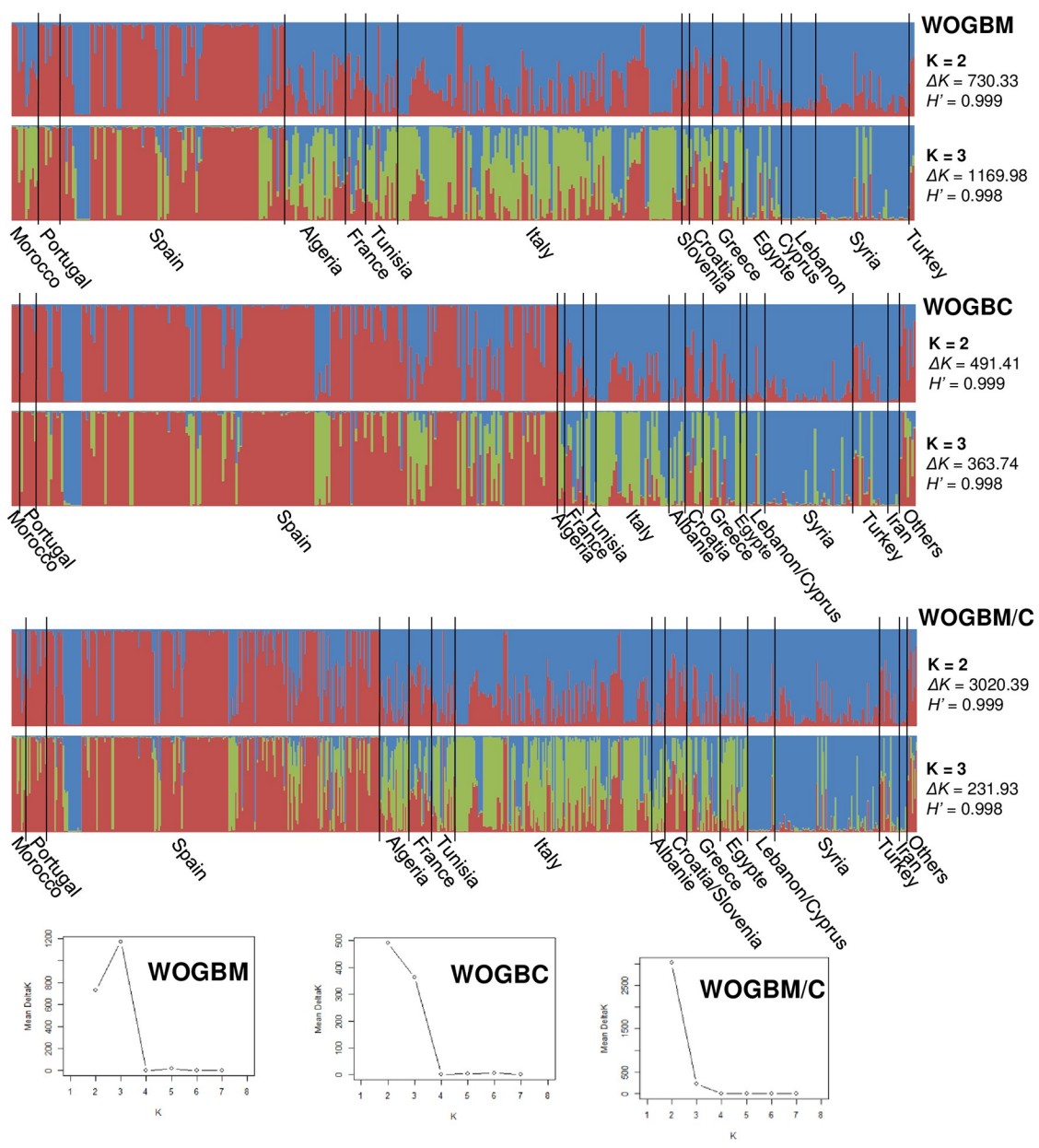

**Fig 1. Inferred structure for K = 2 and 3 within the unique genotypes of the WOGBM (402 genotypes) and WOGBC (400 genotypes) collections and total datasets (672 genotypes; WOGBM/C).** H' represents the similarity coefficient between runs, whereas ΔK represents the ad-hoc measure of Evanno et al. [75].

morphological profiles in WOGBM; and (iii) 662 accessions, including 348 in WOGBC and 314 in WOGBM, as shared profiles with other accessions in both collections, resulting in the identification of 125 different morphological profiles.

**Based on genetic structure.** Based on the whole dataset (672) and data for each collection; 402 for WOGBM and 400 for WOGBC, the genetic structure was examined under the models with K = 2 to 8 clusters. According to ΔK and H', K = 3 were revealed as being the most probable genetic structure model for WOGBM, as previously reported by El Bakkali et al. [23] (ΔK = 1169.98 and H' = 0.998), whereas for WOGBC, K = 2 followed by K = 3 were revealed as being the most probable genetic structure models (Fig 1). With all 672 genotypes, the K = 2

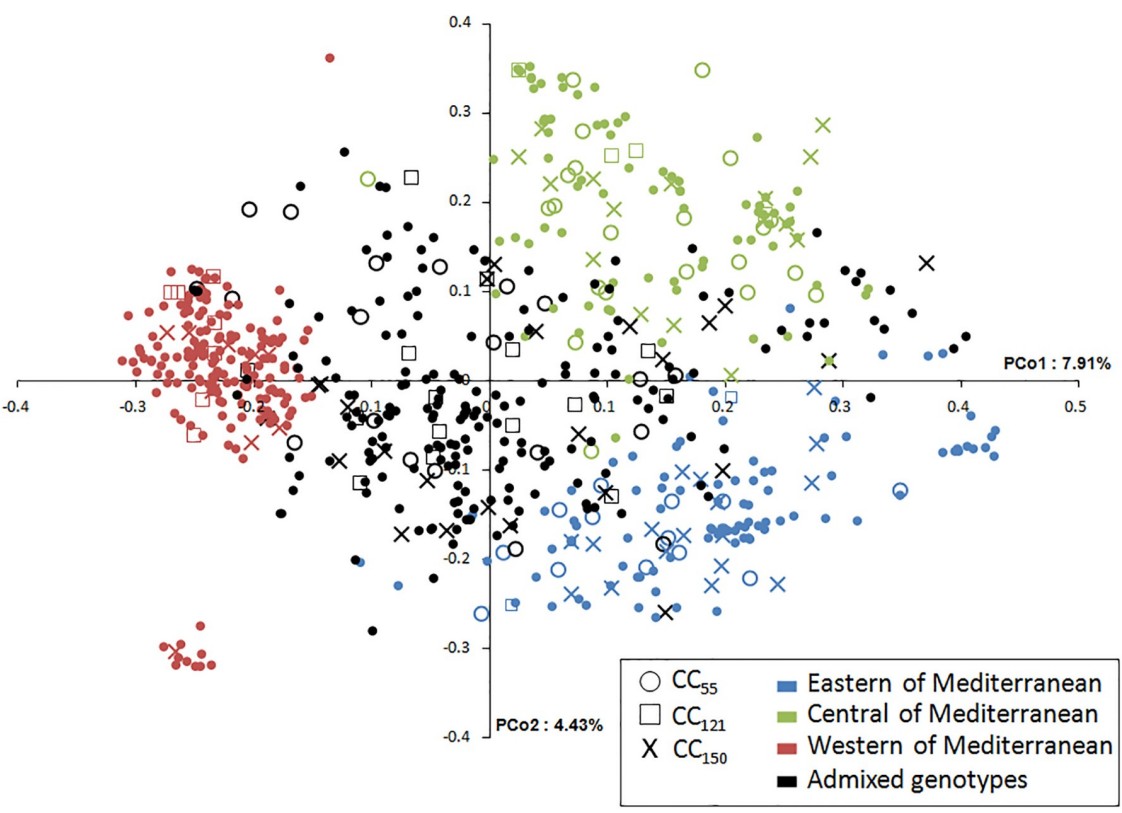

**Fig 2. Two-dimensional distribution of the principal coordinate analysis (PCoA) for both collections and nested core collection of 55, 121 and 150 sample sizes based on SSR markers.** Colors indicated the three gene pools with membership probabilities of Q> 0.8 (eastern, western and central Mediterranean Basin). Nested core collections span the range of all the genotypes among the three gene pools.

model was revealed as being the most probable genetic structure model (ΔK = 3020.39 and H' = 0.999) followed by K = 3 clusters (ΔK = 231.93 and H' = 0.999; Fig 1). Most olive genotypes from Morocco, Portugal and Spain were distinguished from other Mediterranean olives, whereas genotypes from France, Algeria, Albania, Tunisia, Italy, Slovenia, Croatia, Greece and Egypt were mostly assigned as a second cluster (named central Mediterranean) distinguished from the third cluster, which included genotypes from the eastern Mediterranean region. A clear distinction between the three clusters was observed when plotting the 672 genotypes, with membership probabilities of Q< 0.8, using PCoA (Fig 2).

## Cultivar identification process in both WOGBs

Cultivar identification in WOGBM was carried out using nuclear markers and morphological traits as a complement. For accessions with no endocarp trait data, SSR markers were used as the main criteria to identify cultivars and therefore the cultivar identification process was suspended, as mentioned in S1 Table (pending identification). We defined individual cultivars as having accessions with the same molecular and morphological profiles. Molecular variants detected for any identified cultivar were considered in case of minimal differences in SSR profiles between accessions (1 to 4 mismatched alleles) despite similar morphological profiles. We confirmed the varieties authenticated by Trujillo et al. [21] and propose a new set of authenticated varieties on the basis of their matching molecular and morphological profiles with varieties within WOGBC. Detected errors in WOGBM (or mislabeling errors) were considered if

the accession did not match its putative cultivar identified in WOGBC and by considering the cultivar name and area of cultivation. Synonymous cases were considered if the identified cultivars displayed similar (or close) SSR and morphological profiles while having the same or close area of cultivation within and/or between collections. Similarly, homonymy cases were considered if the identified cultivars displayed different SSR and morphological profiles while also considering the area of cultivation within and/or between collections.

**Cultivar identification and authentication processes.** Using endocarp traits as a complement to SSR markers, a total of 535 cultivars were identified in both collections in which 126 cultivars were shared (Table 1 and S1 Table). In addition to the four cultivar pairs identified by Trujillo et al. [21] in WOGBC which showed an identical or nearly identical SSR profile but presented a distinct morphological profile, i.e. "Chemlali-744"/"Chetoui", "Azulejo"/"Manzanilla Cacereña", "Cordovil Castelo Blanco"/"Verdial de Badajoz" and "Zarza"/"Lechin de Sevilla", we identified 8 other cultivar pairs showing similar patterns, i.e. "Crnica-399"/"Crnica", "Ronde de la Ménara"/"Picholine Marocaine", "Jabaluna"/"Escarabajuelo de Úbeda", "Meski"/"Fouji vert"/"Besbessi", "Oblica"/"Lumbardeska", "Beldi"/"Giarfara", "Gremigno di Fauglia"/"Caninese" and "Sourani Red"/"Beladi"/"Beladi-577" (S1 and S4 Tables). Otherwise, between-collection discrepancies were noted regarding the accession names and cultivar identification process. For instance, "Meski" cultivars from Tunisia, "Crnica" from Croatia and "Lentisca" from Spain in WOGBC were found to differ from those in WOGBM (S1 Table).

Finally, we were able to authenticate a total of 120 cultivars in WOGBM. Regarding WOGBC, 110 were previously authenticated from the panel of 200 cultivars described by Trujillo et al. [21], including "Picholine marocaine" from Morocco, "Amargoso" from Spain and "Zaity" from Syria (S1 and S4 Tables). Moreover, 10 new cultivars in WOGBM that were not previously authenticated by Trujillo et al. [21] were found to be authentic as they displayed SSR and morphological profiles similar to those in WOGBC. For instance, "Chemlal de Kabilye" cultivars from Algeria and "Chalkidikis" and "Kolybada" from Greece were authenticated in the current study as they showed similar SSR and morphological profiles in both collections (S1 and S4 Tables).

**Plantation mislabeling and molecular variants.** The identification process allowed identification of 79 cases of mislabeled plantations in WOGBM compared to 120 in WOGBC (S1 Table). For instance, "Chemlali" accessions from Tunisia (MAR00301 and MAR00296) were detected as plantation errors as they showed profiles similar to those of well-known "Koroneiki" and "Arbequina" cultivars, respectively, and different from the profile of "Chemlali" (COR000744) in WOGBC (S1 Table). Similarly, the "Varudo" (MAR00275) accession was identified as differing from its putative cultivar in WOGBC. Moreover, 228 genotypes were considered to be molecular variants of 74 cultivars (1 to 4 dissimilar alleles) because no endocarp morphological differences were observed between them (Fig 3, S6 Table). The highest number of cultivars showing molecular variants was observed for Spanish cultivars with 29 (92 SSR profiles) followed by Italian and Syrian cultivars with 20 (66 SSR profiles) and 9 cultivars (25 SSR profiles), respectively. These 74 cultivars with molecular variants were classified as follows: (i) 13 cultivars observed only in WOGBM with 37 SSR profiles, (ii) 9 cultivars observed only in WOGBC with 21 SSR profiles and (iii) 52 cultivars shared between the two collections with a total of 170 SSR profiles (Fig 3, S6 Table).

**Synonymy and homonymy cases.** A total of 175 possible cases of synonyms among 78 cultivars were identified in this study. We counted 79 cases considered as newly observed in WOGBM and 96 previously described (S7 Table). For instance, a new synonymous group was identified that included four names for the "Picholine marocaine" cultivar formed by the "Hamrani" (MAR00534) accessions from Morocco, "Aghenfas" (MAR00523) and "Limli"

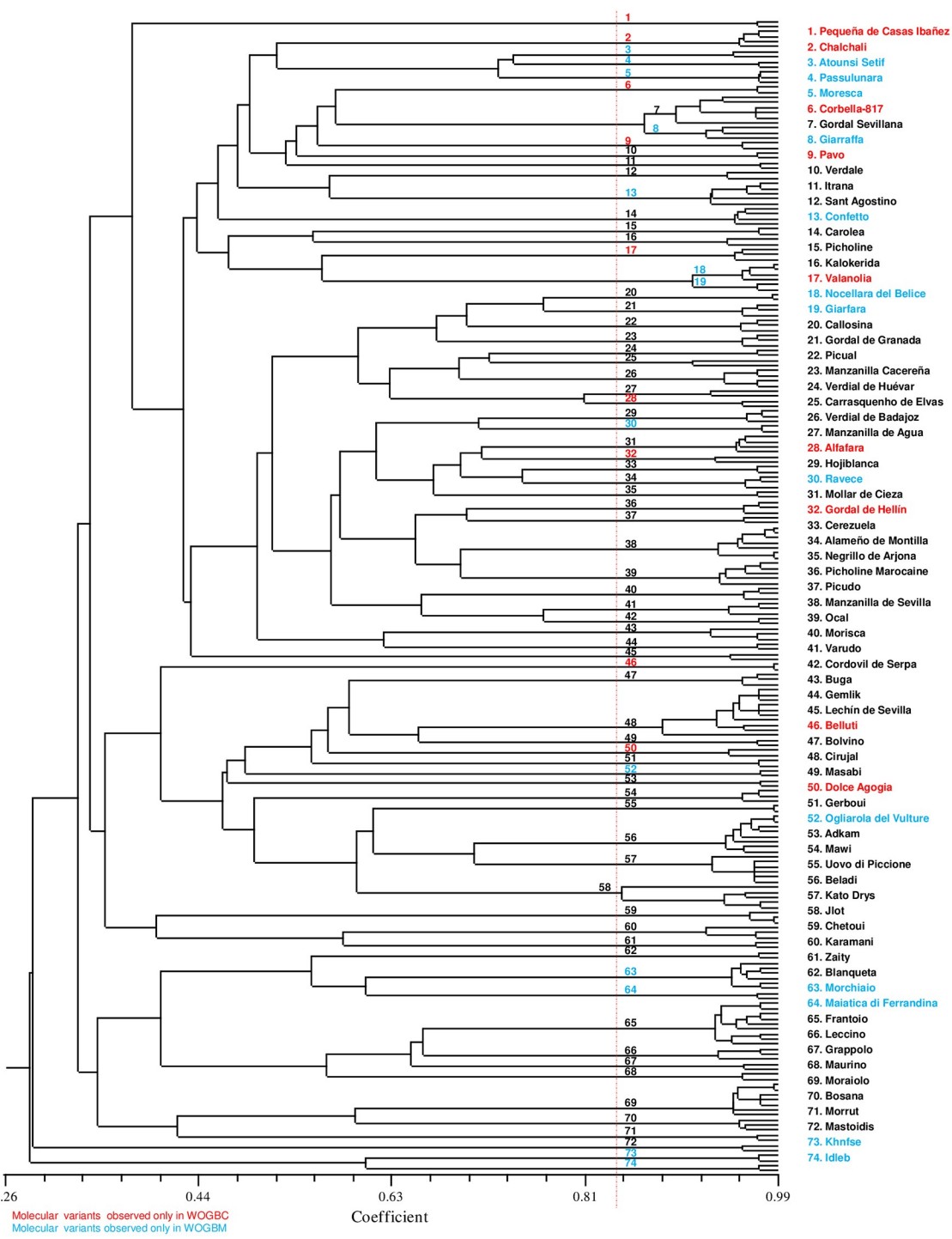

**Fig 3. Dendrogram based on the UPGMA method and Dice similarity index of 74 different cultivars showing molecular variants.** Numbers indicate groups of molecular variants of each cultivar as indicated on the right. Colors indicate the collections in which molecular variants were observed; red color in WOGBC, blue color in WOGBM and black color sharing between the two collections.

(MAR00442) from Algeria and "Sinawy" (MAR00498) from Egypt, and a second group of four names for the "Frantoio" cultivar formed by the "Arancino" (MAR0028), "Puntino" (MAR0054) and "San Lazzaro" (MAR0065) accessions from Italy and "Cailletier" (MAR00190) from France (S7 Table). Moreover, 39 cases of homonyms in WOGBM and 36 in WOGBC were identified among a total of 60 in both collections, thus accounting for 179 identified cultivars, e.g. the "Lentisca" denomination, which included three different cultivars, including "Lentisca" and "Lentisca-244" from Spain and "Lentisca-206" from Portugal. Similarly, the "Corsicana" denomination included two cultivars from Italy, i.e. "Corsicana da Mensa" and "Corsicana da Olio" (S8 Table).

## Core collection sampling

According to the two-step method proposed by El Bakkali et al. [23], 110 individuals (16.3%) were necessary to capture the 407 alleles in the whole dataset using the MSTRAT program (672 genotypes). The CORE HUNTER program was run at 8.2% sample size (half of the initial sample size) using the 'Sh strategy' to sample a primary core collection (CC$_{55}$; S9 Table). The 55 entries thus allowed capture of 328 alleles (80.6%), and only 7 genotypes in common between both collections, while the others were from either WOGBM (33 genotypes) or WOGBC (17). No genotype assigned to western gene pools was selected in the CC$_{55}$ core collection (Table 4).

The primary CC$_{55}$ was used as a kernel with MSTRAT to capture the remaining alleles. Hence, a total of 121 entries (CC$_{121}$; 18%) were sufficient to capture the total diversity and 50 core collections with 121 entries were generated using MSTRAT (S9 Table, S4 Fig). No differences in the Nei diversity index (He) were observed in 50 independent runs. In addition to the 55 genotypes used as a kernel, in all 50 independent runs 32 genotypes were found to be carrying unique alleles, while a combination of 34 complement genotypes could be selected among a panel of 131 genotypes to capture the total diversity (S9 Table). Moreover, 29 varieties which were not selected in the 50 repetitions generated were added, including the most cultivated ones, to account for the morphological trait variability. These 29 varieties could be added to CC$_{121}$ as supplementary genotypes to propose a final core collection of 150 entries (CC$_{150}$; S9 Table).

One core collection CC$_{121}$ was arbitrarily selected in addition to the 29 cultivars added. The 150 genotypes sampled were found to belong to 16 different countries among the 23 analyzed herein (Table 4). Only 39 genotypes shared between the two WOGBs among the 150 genotypes of CC$_{150}$ were selected while the others were either from WOGBM (69 genotypes) or WOGBC (42) and only 16 genotypes were assigned to western genetic clusters compared to 42 and 31

**Table 4. Summary of the nested core collections at different sample sizes compared to the whole dataset.** Number of alleles (Na), Marrakech (M), Cordoba (C).

| N. genotypes | WOGB | | | Na | N. of SSR profiles | N. of morph. Profiles | N. of countries | Genetic clusters | | | |
|---|---|---|---|---|---|---|---|---|---|---|---|
| | M | C | M & C | | | | | Western[1] | Central[1] | Eastern[1] | Admixture[2] |
| 55 | 33 | 15 | 7 | 328 | 55 | 49 | 11 | | 22 | 13 | 20 |
| 121* | 67 | 40 | 14 | 407 | 121 | 95 | 16 | 9 | 38 | 30 | 44 |
| 150 | 69 | 42 | 39 | 407 | 150 | 120 | 16 | 16 | 42 | 31 | 61 |
| 672 | 272 | 270 | 130 | 407 | 672 | 371 | 23 | 183 | 118 | 128 | 243 |

*One core collection was arbitrary selected among the 50 generated using Mstrat (S9 Table).

[1]as identified by Structure program with a membership probability of Q ≥ 0.8

[2]as identified by Structure program with a membership probability of Q < 0.8

that were assigned to central and eastern genetic clusters, respectively. The selected 150 entries were plotted according to genetic cluster assignations using PCoA. The results revealed that the genotypes sampled spanned the entire range of genotypes in the whole dataset among the three genetic clusters (Fig 2).

## Discussion

The main purpose of this study was to develop a consensus database for Mediterranean olive cultivars based jointly on molecular markers and endocarp traits to serve as an efficient tool for scientists conducting research on breeding and adaptation, as well as other users of local genetic resources. Germplasm banks are crucial for breeding programs as they offer variability in target agronomic traits and allelic variation in genes linked to these traits. Development of reference standards and a well-defined nomenclature system free of homonymy, synonymy and molecular variants is essential for making effective use of olive genetic resources. Here we discuss our results based on: (i) the efficiency of the joint use of molecular markers and endo-carp descriptors for olive cultivar characterization; (ii) the relevance of available genetic resources in worldwide collections; and (iii) the importance of this information and of estab-lishing a core collection for the scientific community and other users of olive genetic resources.

### Combining SSR markers and endocarp descriptors to efficiently characterize olive cultivars

Many studies have revealed the efficiency of microsatellites markers for characterizing olive cultivars [14,17,21,22,51,52,63,64]. The 20 SSR markers used here were selected based on their high polymorphism, clear amplification and reproducible patterns, as reported by many authors [21–23,51,52]. Among the 33 SSR loci used by Trujillo et al. [21] for WOGBC genotyp-ing, only 17 were able to discriminate the 411 identified genotypes, while 14 loci were in com-mon with the 20 loci used in the current study. Moreover, eight of the 20 loci were used by Sarri et al. [51] to discriminate between 118 cultivars, whereas 11 were selected among 37 SSR loci by Baldoni et al. [52] as a consensus list of microsatellites recommended for genotyping 77 olive cultivars.

The 20 SSR markers used in the present study were able to discriminate 400 genotypes among the 411 genotypes identified in WOGBC using 33 SSR markers. The 11 genotypes not identified here corresponded to 15 olive accessions, most of which Trujillo et al. [21] consid-ered were molecular variants and/or synonyms of well-known cultivars, e.g."Alameño de Marchena" (COR000254) for "Picholine Marocaine". Only two exceptions were noted for four different cultivars displaying distinct morphological traits: "Zarza" (COR000038)/"Lechin de Sevilla" (COR000005) and "Pulazeqin" (COR001085)/"Itrana" (COR000068), which were revealed to be similar based on 20 instead of 33 loci. We therefore noted that using a lower number of loci did not diminish the discrimination power as only 15 out of 537 profiles were grouped.

Characterization of large collections with a high number of loci is costly. Generating a sub-set of SSR markers able to discriminate all cultivars is thus essential to enable olive researchers to mine local olive diversity relative to known cultivars. The first step would be to eliminate duplicate samples, which could be done using a minimum of SSR loci. Here we aimed to develop a minimum SSR locus subset by exploring both collections. Among the 20 SSR loci used, we identified 3 and 6 markers that could discriminate 77% and 94% of 672 genotypes present in both WOGBs, respectively. Surprisingly, Trujillo et al. [21] proposed a set of 5 and 10 markers to distinguish 79% and 93% of cultivars present in WOGBC, respectively. By

focusing on the 6 markers proposed here, five (UDO-43, DCA04, DCA09, DCA16, and GAPU103) were in common with the 10 proposed by Trujillo et al. [21], which was sufficient to distinguish 91.5% of genotypes (615 among 672). This set of five markers has been widely used in molecular characterization, including parentage analysis [53].

Molecular characterization was complemented by using 11 morphological descriptors related to endocarp traits in both collections. Endocarp traits are still considered to be the most discriminating and stable olive morphological traits as they are not highly influenced by environmental conditions, while being easily and quickly evaluated, even in the field. More-over, the endocarp could be conserved in the long term and used as a reference. Endocarp descriptions have thus been widely used for olive cultivar identification and for clarifying domestication and diversification processes involving stones at archeological sites [26,27,90–93]. However, we used SSR genotyping for a first classification of the different genotypes complemented by considering the endocarp traits: (i) the discrimination power of SSR (672 molecular profiles) compared to endocarp traits (371 morphological profiles); (ii) scoring dis-crepancies regarding endocarp traits due to their qualitative features, which can lead to confu-sion and misclassification of cultivars between observers; and (iii) phenotypic changes due to genetic differences might be expressed in organs other than endocarp (fruit, leaves, etc.). How-ever, endocarp traits are still very useful when considering genetically similar/close cultivars revealed by SSR markers, e.g. "Zara"/"Lechin de Sevilla", "Azulejo"/"Manzanilla Cacereña", "Fouji vert"/"Besbassi", "Olivastra di Montalcino"/"Mortellino" and "Pulazeqin"/"Itrana".

Molecular variants have been regularly reported in olive cultivars [21,28–31], relict wild olive trees in the Hoggar Mountains [94] and even in other fruit crops such as grapevine [31,95] and fig [32]. This slight allelic variation has been noted in olive cultivars that are widely grown in several olive growing areas, e.g. "Frantoio" (Italy), "Picholine Marocaine" (Morocco) and "Picual" (Spain). This has also been observed for ancient cultivars grown during several periods throughout history, e.g. "Picholine Marocaine" [30,96], "Cobrançosa" (Portugal) [29], "Manzanilla de Sevilla" (Spain) [97], "Gemlik" (Turkey) [98] and "Carolea" (Italy) [99]. These cultivars have undergone massive clonal propagation on a spatial or temporal scale, or both. It can thus be assumed that the massive clonal propagation process can lead to slight allelic varia-tions, especially in the case of SSR markers. Indeed, these markers are considered to be muta-tional hotspots due to the presence of short repetitive units [100], and most variations occur at loci with dinucleotides and abundant GA repeat units which are more susceptible to mutations and slippage [101]. Hence, 10 amongst the 17 loci showing intra-varietal molecular variations were noted for SSR loci with abundant GA repeat units and in cultivars which are massively clonally propagated (Fig 3, S10 Table).

Identifying a single SSR profile as a molecular variant of a known cultivar is still challenging as no consensus approach has been proposed to clarify this issue. Here we proposed 228 geno-types as molecular variants of 74 cultivars showing light allelic variations (Fig 3, S6 Table). We considered that a threshold of less than 4 mismatched alleles among the set of 20 SSR loci would be strong enough to classify close accessions displaying similar endocarp traits within a single cultivar. As the allelic variation could be extended to up to six mismatched alleles (S1 Fig), we adopted a conservative but robust approach to identify these molecular variants. Sev-eral observations confirmed the relevance of our approach: (i) pairwise comparisons between the two WOGBs revealed that for WOGBM and WOGBC only 0.68% and 0.42% pairs, respec-tively, represented closely related genotypes with 1 to 4 distinct alleles, whereas the remaining pairs were mostly distinguished by 8 to 39 distinct alleles (S1 Fig); (ii) for all genotypes show-ing less than 4 mismatched alleles, similar endocarp profiles were observed, whereas significant differences were revealed at 5 mismatched alleles, e.g. "Morchiaio"/"Razzaio" and "Manto-nica"/"Cirujal"; (iii) most genotypes showing less than 4 dissimilar alleles were identified

within the same country (e.g. "Kato Drys" cultivar from Cyprus) or among closely located countries (e.g. "Morchiaio" cultivar from Italy and Slovenia), indicating that massive clonal propagation has been under way in large growing areas, thus leading to slight molecular variation; and (iv) Trujillo et al. [21], based on 33 SSR markers, considered that all accessions with a similarity index of 0.99 to 0.91, corresponding to 1 to 5 mismatched alleles, were molecular variants.

## Identifying olive cultivars

With more than 1,200 olive varieties from around the Mediterranean Basin found in almost 100 *ex-situ* collections [13], two major goals have to be fulfilled: (i) characterization to eliminate mislabeling cases and identify synonymy and homonymy cases, and (ii) authentication of cultivars to ensure that accessions in one collection match true-to-type cultivars. However, most studies have been focused on characterization of *ex-situ* collections despite the fact that cultivar authentication should be a pre-requisite for exchanging material among scientists and nurseries. Trujillo et al. [21] defined authenticated cultivars based on their identity with respect to endocarp control samples from their original growing areas and DNA control samples. Hence, they were able to authenticate 200 cultivars by comparing endocarps with those of the same cultivar from the countries of origin (172 cultivars) and both endocarp and control DNA samples (28 cultivars). Even though the combination of molecular and morphological characterization was applied only to 28 out of 200 cultivars, the authentication approach used by Trujillo et al. [21] could be considered efficient since it targeted known cultivars from Spanish germplasm which represented 66% of the authenticated cultivars. However, for local and minor cultivars that are less known but especially present in limited geographic areas, the authentication process should systematically include both molecular and morphological characterization, as previously proposed for French olive germplasm [47]. Indeed, these authors defined a reference genotype for one cultivar when at least three olive trees grouped under the same denomination while presenting similar morphological traits and originating from different collections, nurseries and/or orchards display the same molecular pattern. Here, regarding the limits noted in the authentication process, we adopted a conservative strategy by focusing on the panel of 200 cultivars previously authenticated by Trujillo et al. [21]. We were able to authenticate 120 cultivars among the 329 identified in WOGBM, and most of them originated from Spain (81 cultivars; 67.5%). All of these 120 cultivars were previously authenticated in WOGBC by Trujillo et al. [21], except for 10 cultivars with matching molecular and morphological profiles between the two WOGBs.

Within and among accession mislabeling cases could occur at any step during plant establishment in the collection, as reported by many authors with regard to different fruit species [33,47]. Here we identified 79 cases of plantation mislabeling in WOGBM compared to 120 in WOGBC (S1 Table). Using the same approach as Trujillo et al. [21], mislabeling error was assumed when the profile of a known cultivar did not match its putative profile. Hence, when considering identified cultivars based on both molecular markers and morphological traits, Trujillo et al. [21] highlighted that the "Chemlal de Kabylie" (COR000118-Algeria), "Zaity" (COR000788-Syria), "Adkam" (COR001038-Syria) and "Aggezi Shami" (COR000723-Egypt) accessions were "Frantoio" cultivar mislabeling errors. Similarly, here for the same cultivar, i.e. "Frantoio", we highlighted three accessions as mislabeling errors: "Fakhfoukha" (MAR00533-Morocco), "Jlot" (MAR00587-Lebanon) and "Beladi" (MAR00583-Lebanon). This highlights the mislabeling issues that may arise as a result of broad diffusion of clonally propagated known cultivars such as "Frantoio".

The cultivar identification process was helpful for describing homonymy and synonymy cases in both collections. A total of 39 homonymy cases in WOGBM and 36 in WOGBC were identified among a total of 60 in the two collections involving a total of 179 identified cultivars, e.g. the Lentisca denomination which encompassed three distinct genotypes. Many authors have reported homonymy cases in olive [14,21,22,27,102,103]. Here we noted that homonymy cases mostly involved well known and widely cropped cultivars such as "Azeradj" (Algeria), "Cornicabra", "Manzanilla" and "Picual" (Spain), "Toffahi" (Egypt and Syria). This reflected a farming strategy of known cultivar appropriation whereby distinct genotypes were pooled under a single denomination.

Otherwise, synonymy is likely the result of plant material dissemination and of traditional olive vegetative propagation practices. Such practices led to complex relationships among cultivars. We identified 175 synonymy cases among 78 cultivars in both collections (e.g. the "Gordal Sevillana" cultivar from Spain was synonymous with "Santa Caterina" from Italy). We highlighted 88 new synonyms in WOGBM or shared between the two collections including not previously described cultivars such as "Kiti", "Meniko" and "Peristerona" from Cyprus as synonyms of "Kato Drys", whereas 87 had already been described by other authors (e.g. "Olivastra di Montalcino" as a synonym of "Olivastra Seggianese" described by Cimato et al. [104] (S7 Table). Many synonym cases identified in WOGBM were in agreement with those obtained by Trujillo et al. [21] in WOGBC as they showed similar morphological and molecular profiles in both collections, e.g. "Sigoise"/"Alameño de Marchena"/"Haouzia" for "Picholine marocaine", thus showcasing the power and efficiency of tools used to characterize and compare the two collections. The high number of synonymy cases observed in WOGBM compared to WOGBC could be explained by two factors. First, accessions received from many countries during the establishment of WOGBM were characterized *in-situ* using only morphological traits (fruit, endocarp, leaves, etc.), which could generate confusion compared to characterization based on the combination of morphological descriptors and molecular markers. For instance, accessions from Cyprus, Syria and Slovenia in which 31, 70 and 10 accessions, respectively, were analyzed and only 4 (13%), 42 (60%) and 3 (30%) genotypes were identified, respectively (Table 1). Conversely, analyzed accessions from Spain were found to be distinct due to previous analyses using molecular markers prior to WOGBM establishment [46]. Second, the varietal composition in the two collections contrasted with at least 50% of germplasm in WOGBC originated from Spain. This could be explained by the fact that most of the synonymy cases identified by Trujillo et al. [21] concerned Spanish cultivars (17 cultivars among 30 cultivars). Here, the highest number of synonyms was revealed for three cultivars that are widely cultivated throughout the Mediterranean Basin, with 13 cases for each one: "Beladi", "Frantoio" and "Picholine marocaine" (S7 Table). Most synonymy cases were observed within one country or among closely located countries.

Identifying synonymy cases is essential for managing *ex-situ* collections by eliminating denomination redundancy. The synonymy cases identified in the present study could be a focus of interest for the olive research community, and we are aware that further investigations with large samples and comparisons of several control samples are required using both endocarp and DNA control samples.

In perennial clonally-propagated fruit species, a cultivar is defined as a group of similar plants that have been selected for one or more interesting characters that are distinct, uniform and stable. It is thus not surprising to observe cases of molecular variants, synonymy and homonymy within cultivars resulting from vegetative propagation and long-term diffusion of domesticated olive via farming practices. Here, within the two collections, we identified a total of 539 cultivars among 672 genotypes, 210 of which are considered as authenticated cultivars. We also defined a set of six SSR makers as efficient tools for discriminating almost all diversity

present in both collections. The database generated and the approach used in our study could be applied to compare and harmonize more collections at national (Italy-Cosenza, [14]; Turkey-Izmir, [15]; Greece-Chania, [16]; France-Porquerolles, [47]) and international (USA-Davis, [17]; Argentina-Mendoza, [18]) levels. Setting up an international consortium for the identification, authentication and cataloguing of more than 1,200 cultivars across the Mediterranean Basin, under a common protocol with the six SSR loci selected in the present study and 11 endocarp traits is a necessary step. The expected results would consolidate partnership cooperation between communities of scientists conducting research on olive breeding and other users of olive genetic resources. Our study will enhance the establishment of the third worldwide collection in the eastern Mediterranean Basin (Izmir-Turkey) and will help to update and extend the Olea database [11], which is the most comprehensive global olive science portal to date.

The identification and authentication of cultivars in the two largest worldwide olive germplasm collections will ensure sustainable use of local genetic resources. In fact, studies on the resilience of true-to-type cultivars in different environments using germplasm collections across the IOOC network will guarantee the future of the crop under different climate change scenarii. The use of true-to-type and healthy plant material by commercial nurseries and by farmers for selected local cultivars will promote the use of authentic cultivars through protected designations of origin (PDO).

## The importance of pooling the two collections

The two WOGB collections have two distinct histories. WOGBC, as the top olive germplasm bank in the world, was established in 1970 and contains 499 accessions from 21 countries, most of which originated from Spain (56%) [21]. WOGBM, as the second ranking olive germplasm bank, was established 33 years later and contains accessions from 14 countries. Compared to WOGBC, WOGBM was set up in a scientific setting with more knowledge available about the plant material and previously characterized genetic resources were introduced from each Mediterranean country [22]. Therefore, the Marrakech collection has a more balanced representation of accessions from Mediterranean olive growing countries such as Morocco, Algeria, Tunisia, Egypt and Cyprus, whereas the Cordoba collection has accessions from Albania, Turkey and Iran that are not included in the Marrakech collection. Hence, the two collections are complementary and not duplicated, as supported by the following observations. First, only 178 accession names among 713 are shared between the two collections. Only 130 genotypes (20%) out of 672 are in common, with a dominance of Spanish olive germplasm. Forty-eight discrepancy cases were identified, including Meski, Lentisca and Varudo, which could mainly be explained by mislabeling errors during germplasm propagation and/or subsequent planting. Second, the high proportion of accessions and therefore genotypes and cultivars specific to each collection was noted due to the contrasted varietal composition of each collection. Third, the genetic structure analysis revealed that the most probable genetic structure models are K = 2 and K = 3 for the Cordoba and Marrakech collections, respectively, while with all of the 672 genotypes, the K = 2 model was found to be the most relevant genetic structure model ($\Delta$K = 3020.39 and H' = 0.999). The presence of a high proportion of Spanish germplasm in WOGBC has certainly contributed to the genetic structure pattern observed when pooling the two datasets. However, a high proportion of alleles were shared between the two collections, i.e. 309 alleles (75.9%), despite the contrasted varietal composition between the two collections. Moreover, genetic diversity in terms of allelic richness (Ar) and diversity index (He) were similar and no significant differences were observed between the two collections, as previously reported by Trujillo et al. [21] using 11 SSR loci in common. Hence, the level of genetic

diversity remained stable despite the considerable contrasted varietal composition between collections.

As more than 1,200 olive cultivars have been reported in the Mediterranean Basin [11,37], the current set of cultivars preserved and identified in both collections represented almost 45% of the described cultivars (535 cultivars). Further studies are required to encompass all olive germplasm from the Mediterranean Basin through the inclusion of new local accessions from different countries that are less represented in the two collections, such as French germplasm (more than 150 cultivars [47]), Cosenza-Italy (500 cultivars [14]), Turkey-Izmir (96 cultivars [15]) and Greece-Chania (47 cultivars [16]).

Here we propose a nested core collection based on the two WOGB collections, at three different levels, i.e. 55, 121 and 150 sample sizes. Core collections facilitate the exchange and efficient use of germplasm by providing a set of cultivars representing the overall genetic diversity available in two germplasm banks. We adopted the two-step methodology proposed by El Bakkali et al. [23], which has been found to be the most efficient approach for selecting subsets suitable for genetic association mapping. Our approach was based exclusively on genetic criteria but we believe that the diversity sampled in core collections based on SSR markers only are correlated with other criteria such as morphological and agronomic traits. Compared to WOGBM alone, a high number of cultivars were sampled that captured the total diversity in the two collections, as reported by El Bakkali et al. [23]; 121 vs 94, which could be explained by the contribution of specific alleles present in WOGBC (37 alleles). At the 55 sample size, only 7 genotypes (12.7%) were in common between the two collections, whereas 33 (60%) were revealed to be from WOGBM. Similar proportions were observed at the 121 sample size where 55.3% were from WOGBM and 11.6% were shared between the two collections (Table 4). These observations underline the importance of cultivar variability within WOGBM.

Defining a set of cultivars to serve as a core collection for the two WOGBs and its field assessment will certainly enhance knowledge on the genetic basis of target agronomical traits which is still at early stage. Genetic association mapping using unrelated cultivars is a powerful tool and has been successfully used to identify the genetic basis of many complex traits in plants [105,106]. The proposed core collection will boost insight into the genetic basis of most agronomic and adaptive traits in olive by taking advantage of advances achieved in next generation sequencing (NGS) with the development of high-throughput SNP markers through genotyping by sequencing approaches (GBS) [107,108], while also exploiting the recently released olive reference genome (http://olivegenome.org/) [59,60].

## Conclusion

Intensive olive cropping systems with mechanical harvesting are leading to a reduction in the number of cultivars used in orchards, therefore increasing the risk of genetic erosion. Genetic resources included in germplasm collections represent a major pillar for sustainable use of olive genetic resources and for breeding programs geared towards selecting new resilient cultivars that are better adapted to climate change. Cultivar identification and authentication should thus be compulsory before using olive plant material. Here we first harmonized allele sizes between the two worldwide collections for a set of 20 SSR markers and used 11 endocarp traits as complementary tools for the characterization and identification of olive cultivars. We thus conducted the first in-depth analysis on olive cultivar germplasm. The information and database generated from this study will help manage olive cultivars from the Mediterranean Basin through the launch of a consortium under IOOC supervision. They also represent valuable tools for conducting further studies such as genetic association mapping.

## Supporting information

**S1 Table. Characterization of all olive trees in WOGBM and WOGBC (1091).** Codes in collections, names of accessions and their origins, number of trees analyzed, number of SSR profiles, SSR code, morphological code, name of the identified cultivar and main cultivation area are specified. Identified and authenticated cultivars are mentioned.
(XLSX)

**S2 Table. List of the 47 accessions from WOGBC used for alignment of alleles from both collections with 20 SSR markers.** Each allele size for each locus and each accession in two different genotyping conditions were superimposed.
(DOCX)

**S3 Table. Data for the 672 SSR profiles observed in both WOGBM and WOGBC collections.** Shared genotypes and those specific to each collection are indicated in the WOGB column.
(DOCX)

**S4 Table. List of identified and authenticated cultivars in both collections.** Shared cultivars and specific to each collection are indicated.
(DOCX)

**S5 Table. Summary of genetic parameters of different loci regarding shared genotypes and those specific to each collection.** Number of alleles (Na), expected (He) and observed (Ho) heterozygosity, allelic richness (Ar).
(DOCX)

**S6 Table. Cases of cultivars showing molecular variants with their cultivation area, the number of molecular variants (No.MV) in both collections and the number of distinct alleles.**
(DOCX)

**S7 Table. Cases of synonyms found in the identification process in Marrakech and Cordoba collections.**
(DOCX)

**S8 Table. Cases of homonyms found in the identification process in Marrakech and Cordoba collections.**
(DOCX)

**S9 Table. List of cultivars sampled as nested core collections of 55, 121 and 150 sample sizes.** Cultivar names, origins, SSRs and morphological codes are indicated.
(XLSX)

**S10 Table. SSR markers involved in the description of molecular variants (MV) in the two collections, their repetitive motifs and alleles showing variation compared to reference alleles.**
(DOCX)

**S1 Fig. Frequency distribution of genetic dissimilarity for all pairwise combinations among different genotypes for WOGBM (1) and WOGBC (2).**
(TIF)

**S2 Fig. Pairwise distribution of genetic distances for shared genotypes and specific to each collection.**
(TIF)

**S3 Fig. PCoA plot of 672 olive genotypes identified in both WOGB collections based on SSR data according to the gene pools assignation with membership probabilities of Q> 0.8.**
(TIF)

**S4 Fig. Sampling efficiency to capture total allelic diversity via the M-strategy (M-method) and a random strategy for the whole dataset (672 genotypes) when 55 genotypes, identified by the CoreHunter program, were used as kernels.**
(TIF)

## Acknowledgments

We would like to thank **Luis Rallo, Conception M Diez, Isabel Trujillo, Diego Barranco** for providing us DNA of 47 accessions from WOGB Cordoba and for their kind remarks on an earlier version of the manuscript; **Claire Lanaud** and the **reviewers** for comments on the final version of the manuscript. Molecular analysis was conducted at UMR AGAP Montpellier and we acknowledge the staff of the Genotyping platform. We also acknowledge the **International Olive Oil Council** and **INRA Morocco** for their contribution towards the management of WOGB Marrakech.

## Author Contributions

**Conceptualization:** Ahmed El Bakkali, Bouchaib Khadari.

**Data curation:** Ahmed El Bakkali, Ronan Rivallan, Pierre Mournet.

**Formal analysis:** Ahmed El Bakkali, Bouchaib Khadari.

**Funding acquisition:** Bouchaib Khadari.

**Investigation:** Abdelmajid Moukhli, Hayat Zaher, Lhassane Sikaoui, Bouchaib Khadari.

**Methodology:** Ahmed El Bakkali, Laila Essalouh, Christine Tollon, Ronan Rivallan, Pierre Mournet, Abderrahmane Mekkaoui, Amal Hadidou, Bouchaib Khadari.

**Project administration:** Bouchaib Khadari.

**Resources:** Ronan Rivallan, Abdelmajid Moukhli, Hayat Zaher, Lhassane Sikaoui.

**Supervision:** Bouchaib Khadari.

**Validation:** Ahmed El Bakkali, Bouchaib Khadari.

**Writing – original draft:** Ahmed El Bakkali, Bouchaib Khadari.

**Writing – review & editing:** Laila Essalouh, Pierre Mournet, Abdelmajid Moukhli, Hayat Zaher, Lhassane Sikaoui.

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
