## [Decision Letter · Decision Letter 0]

7 Aug 2019

PONE-D-19-19157

Characterization of Worldwide Olive Germplasm Banks of Marrakech (Morocco) and Córdoba (Spain): towards management and use of olive germplasm in breeding programs

PLOS ONE

Dear Dr KHADARI,

Thank you for submitting your manuscript to PLOS ONE. After careful consideration, we feel that it has merit but does not fully meet PLOS ONE’s publication criteria as it currently stands. Therefore, we invite you to submit a revised version of the manuscript that addresses the points raised during the review process.

Address the various points raised by each of the two reviewers.

We would appreciate receiving your revised manuscript by Sep 21 2019 11:59PM. To enhance the reproducibility of your results, we recommend that if applicable you deposit your laboratory protocols in protocols.io, where a protocol can be assigned its own identifier (DOI) such that it can be cited independently in the future. For instructions see: http://journals.plos.org/plosone/s/submission-guidelines#loc-laboratory-protocols

We look forward to receiving your revised manuscript.

Kind regards,

Randall P. Niedz

Academic Editor

PLOS ONE

Journal Requirements:

Reviewers' comments:

Reviewer's Responses to Questions

**Comments to the Author**

1. Is the manuscript technically sound, and do the data support the conclusions?

Reviewer #1: Yes

Reviewer #2: Yes

2. Has the statistical analysis been performed appropriately and rigorously? 

Reviewer #1: Yes

Reviewer #2: Yes

3. Have the authors made all data underlying the findings in their manuscript fully available?

Reviewer #1: Yes

Reviewer #2: Yes

4. Is the manuscript presented in an intelligible fashion and written in standard English?

Reviewer #1: Yes

Reviewer #2: Yes

5. Review Comments to the Author

Reviewer #1: Dear Editor,

The manuscript entitled: Characterization of Worldwide Olive Germplasm Banks of Marrakech (Morocco) and

Córdoba (Spain): towards management and use of olive germplasm in breeding programs, describe worldwide olive germplasm using morphological and molecular approaches. The article can suitable for olive breeder and useful for extended genetic diversity, which is a main role in plant breeding. the manuscript could be consider for publication in PLOS ONE, but before that require some parts revision as following of:

- The manuscript need to be minor revised for grammatical using native speaker English language because some parts hard understands for readers.

-please prepare high resolution of figures, because the quality is low and some parts no clear in details.

-in Figure 2 and Fig S3, 2-D distribution (PCO) how does it and which of traits (Morphological/molecular data)? The first two PCO accounting 12.34%in total. This is very, when you are established PCO for morphological traits/markers because of different characters for example environmental conditions (Morocco & Spain) you should get high genetic diversity. Unless, some of genotypes have a clone and so express as the same function. This is also expected when we are used molecular data (SSR.RAPID, AFLP, SNP, etc.) with limited primers (for example in your study, 20 SRR markers). These cause that's use markers not comprehensively covered the olive genomes. The authors need to be declared this event and explained.

-One of the important issues in the evaluation and characterization of Plant Germplasm (PG) is their use in plant breeding programs. So knowledge of gene diversity and genetic variation is very important. The authors need to be clarified of it based on Nei & Li (1976; 1979).This is procedure could better helps to identified all the best diversity for selection of genotypes in breeding programs.

- In Figure 3, dendrogram based on UPGMA using Dice coefficient similarity. Why used dice index for clustering? The authors require to analysis Mantel test and so find the best correlation coefficient among (for instance, Jacards, Dice, SM) so we could described our results based on all the best. For stability and consistency analysis of the cluster, bootstrap analysis could better results combined with STRACTURE.

- In Figure 4 the patterns of morphological showed different variation, but we do not see in PCO- analysis! 12.34% is a high!

- Determining the genetic structure of populations is becoming an increasingly important aspect of genetic studies. One of the most frequently used methods is the calculation of F-statistics using an Analysis of Molecular Variance (AMOVA).Linking AMOVA and K-means cluster could better find results in survey of germplasm and is one of significant approaches in plant breeding. Please see AMOVA-Based Clustering of Population Genetic Data (Patrick G. Meirmans, 2012).

Reviewer #2: REFEREE COMMENTS

The manuscript "Characterization of Worldwide Olive Germplasm Banks of Marrakech (Morocco) and Córdoba (Spain): towards management and use of olive germplasm in breeding programs" by Bakkali et al.

is a very interesting paper which aims to discriminate and authenticate 1091 olive accessions belonging to Worldwide Olive Germplasm Bank of Córdoba (WOGBC), Spain, and Marrakech (WOGBM), Morocco, applying 20 microsatellite markers (SSR) and 11 endocarp morphological traits, providing useful information for managing olive germplasm for its preservation, exchange and use in breeding programs.

I recommend publishing this paper with minor revision.

I found very few points to improve, as follows:

Figure are not of good quality, but probably it is a problem of the transferred copy.

Page 2, Line 46: I doubt of these data; please, check it

Page 3, Line 99: On this subject, there are more recent papers worthy to be mentioned:

for SSR analysis

di Rienzo V. et al., 2018. Genetic flow among olive populations within the Mediterranean basin. PeerJ 6:e5260 https://doi.org/10.7717/peerj.5260

Boucheffa S et al., 2017. The coexistence of oleaster and traditional varieties affects genetic diversity and population structure in Algerian olive (Olea europaea) germplasm. (doi: 10.1007/s10722-016-0365-4).

for SNP analysis

D’Agostino, N. D. et al., 2018. GBS-derived SNP catalogue unveiled wide genetic variability and geographical relationships of Italian olive cultivars. Scientific Reports., 1–13. doi:10.1038/s41598-018-34207-y.

Taranto et al., 2018. SNP diversity in an olive germplasm collection. Acta Horticulturae 1199, pp. 27-32. doi: 10.17660/ActaHortic.2018.1199.5

Page 23 Line 917: Table 4 (space)

Page 13 Line 485 On this subject, there are more recent papers worthy to be mentioned:

di Rienzo et al., 2018. The preservation and characterization of Apulian olive germplasm biodiversity. Acta Hortic. 1199, 1–6. doi:10.17660/ActaHortic.2018.1199.

Page 15 line 555

Boucheffa S. et al., 2019. Diversity Assessment of Algerian Wild and Cultivated Olives (Olea europeae L.) by Molecular, Morphological, and Chemical Traits. Eur. J. Lipid Sci. Technol.121, 1800302 DOI: 10.1002/ejlt.201800302.

6. PLOS authors have the option to publish the peer review history of their article (what does this mean?). If published, this will include your full peer review and any attached files.

Reviewer #1: No

Reviewer #2: No

---

## [Author Response · Author response to Decision Letter 0]

23 Sep 2019

Response to reviewers

Reviewer #1: Dear Editor,

The manuscript entitled: Characterization of Worldwide Olive Germplasm Banks of Marrakech (Morocco) and Córdoba (Spain): towards management and use of olive germplasm in breeding programs, describe worldwide olive germplasm using morphological and molecular approaches. The article can suitable for olive breeder and useful for extended genetic diversity, which is a main role in plant breeding. the manuscript could be consider for publication in PLOS ONE, but before that require some parts revision as following of:

- The manuscript need to be minor revised for grammatical using native speaker English language because some parts hard understands for readers.

The article has been reviewed by a professional scientific English mother-tongue reviser (David Manley).

-please prepare high resolution of figures, because the quality is low and some parts no clear in details.

The quality of the figures has been reviewed by using the Preflight Analysis and Conversion Engine (PACE) digital diagnostic tool implemented in PlosOne website

-in Figure 2 and Fig S3, 2-D distribution (PCO) how does it and which of traits (Morphological/molecular data)? The first two PCO accounting 12.34% in total. This is very, when you are established PCO for morphological traits/markers because of different characters for example environmental conditions (Morocco & Spain) you should get high genetic diversity. Unless, some of genotypes have a clone and so express as the same function. This is also expected when we are used molecular data (SSR.RAPD, AFLP, SNP, etc.) with limited primers (for example in your study, 20 SRR markers). These cause that's use markers not comprehensively covered the olive genomes. The authors need to be declared this event and explained.

For the PCoA in Figure 2, we used only SSR data. Due to the high genetic diversity within both collections and to high number of alleles par locus (average = 20.35, Table 2), the total variation is described by the N-1 axis, where N is the number of alleles. In the figure below, more than 100 axes are presented with the inertia proportion exceeding 0.1%. The first two axes explained the highest variation, i.e. 12.34%. This pattern is mostly observed in studies based on SSR (Diez et al., 2014, New Phytologist; Emanuelli et al., 2013; BMC Plant Biology; Singh et al., 2013, Plos One). 

-One of the important issues in the evaluation and characterization of Plant Germplasm (PG) is their use in plant breeding programs. So knowledge of gene diversity and genetic variation is very important. The authors need to be clarified of it based on Nei & Li (1976; 1979).This is procedure could better helps to identified all the best diversity for selection of genotypes in breeding programs.

We agree on the fact that knowledge of gene diversity and genetic variation is very important, especially in breeding programs. We examined the organization of genetic diversity using two approaches, the first one was the model-based clustering method implemented in Structure (Figure 1) and the second one was PCoA based on simple matching coefficient (Figure 2). We did not use clustering analysis based on Nei & Li (1976; 1979). As additional information to the reviewer, we used the DAPC approach implemented in Adegenet package in R to classify the 672 olive genotypes in different groups and then we constructed one dendrogram based on NJ clustering and Fst for each pairwise group of genotypes (see below for details). We obtained similar pattern as Structure and PCoA analyses.

- In Figure 3, dendrogram based on UPGMA using Dice coefficient similarity. Why used dice index for clustering? The authors require to analysis Mantel test and so find the best correlation coefficient among (for instance, Jacards, Dice, SM) so we could described our results based on all the best. For stability and consistency analysis of the cluster, bootstrap analysis could better results combined with STRUCTURE.

For the dendrogram in Figure 3, we used the UPGMA method and Dice similarity index to classify the set of 74 different cultivars showing molecular variants. For each of these cultivars, their molecular variants are distinguished by only one to four dissimilar alleles. Our aim here was not to analyze the genetic relationships between all of the 672 olive genotypes identified in both collections, but we focused on the analysis of molecular variants within each of the 74 cultivars with presumed somatic mutations. We examined the genetic relationships between the 672 olive genotypes using the Structure and PCoA approaches described in Figures 1 and 2. Furthermore, we proposed additional analyses based on the clustering approach described above. 

- In Figure 4 the patterns of morphological showed different variation, but we do not see in PCO- analysis! 12.34% is a high!

We choose to remove the figure 4 from our manuscript as it doesn’t show clearly the similarity in the endocarp traits between synonymous cases of Picholine marocaine (Limli, aghenfas…) and Frantoio (San Lazzero, Cailletier …) cultivars. The PCoA was performed based only on SSR markers.

- Determining the genetic structure of populations is becoming an increasingly important aspect of genetic studies. One of the most frequently used methods is the calculation of F-statistics using an Analysis of Molecular Variance (AMOVA). Linking AMOVA and K-means cluster could better find results in survey of germplasm and is one of significant approaches in plant breeding. Please see AMOVA-Based Clustering of Population Genetic Data (Patrick G. Meirmans, 2012).

Upon the reviewer’s request, we performed an additional analysis to identify the number of clusters using the K-means clustering approach, as implemented in the adegenet package in the R environment. The most probable group was detected at 14 (figure a below). Based on these 14 groups, Pairwise Fst differentiation was calculated and ploted using NJ method (see figure b below). The 14 identified groups were classified based on pairwise Fst into three major clusters (west, east and centre) with a high bootstrap. such genetic pattern is similar to Bayesian model approach (Structure analysis) and PCoA analysis.

Reviewer #2: REFEREE COMMENTS

The manuscript "Characterization of Worldwide Olive Germplasm Banks of Marrakech (Morocco) and Córdoba (Spain): towards management and use of olive germplasm in breeding programs" by Bakkali et al. is a very interesting paper which aims to discriminate and authenticate 1091 olive accessions belonging to Worldwide Olive Germplasm Bank of Córdoba (WOGBC), Spain, and Marrakech (WOGBM), Morocco, applying 20 microsatellite markers (SSR) and 11 endocarp morphological traits, providing useful information for managing olive germplasm for its preservation, exchange and use in breeding programs.

I recommend publishing this paper with minor revision.

I found very few points to improve, as follows:

Figure are not of good quality, but probably it is a problem of the transferred copy.

OK, the quality of the figures has been reviewed

Page 2, Line 46: I doubt of these data; please, check it

The data were updated based on IOOC and FAO statistics (most recent data from 2018)

Page 3, Line 99: On this subject, there are more recent papers worthy to be mentioned:

for SSR analysis 

di Rienzo V. et al., 2018.. PeerJ 6:e5260 https://doi.org/10.7717/peerj.5260

Boucheffa S et al., 2017. The coexistence of oleaster and traditional varieties affects genetic diversity and population structure in Algerian olive (Olea europaea) germplasm. (doi: 10.1007/s10722-016-0365-4).

for SNP analysis

D’Agostino, N. D. et al., 2018. GBS-derived SNP catalogue unveiled wide genetic variability and geographical relationships of Italian olive cultivars. Scientific Reports., 1–13. doi:10.1038/s41598-018-34207-y.

Taranto et al., 2018. SNP diversity in an olive germplasm collection. Acta Horticulturae 1199, pp. 27-32. doi: 10.17660/ActaHortic.2018.1199.5

These references were included in the final version of the manuscript.

Page 23 Line 917: Table 4 (space)

This comment has been taken into consideration in the final version

Page 15 line 555

Boucheffa S. et al., 2019. Diversity Assessment of Algerian Wild and Cultivated Olives (Olea europeae L.) by Molecular, Morphological, and Chemical Traits. Eur. J. Lipid Sci. Technol.121, 1800302 DOI: 10.1002/ejlt.201800302.

This reference has been Taken into consideration in the final versión.

---

## [Editor Report · Decision Letter 1]

27 Sep 2019

Characterization of Worldwide Olive Germplasm Banks of Marrakech (Morocco) and Córdoba (Spain): towards management and use of olive germplasm in breeding programs

PONE-D-19-19157R1

Dear Dr. KHADARI,

We are pleased to inform you that your manuscript has been judged scientifically suitable for publication and will be formally accepted for publication once it complies with all outstanding technical requirements.

With kind regards,

Randall P. Niedz

Academic Editor

PLOS ONE
---

## [Editor Report · Acceptance letter]

8 Oct 2019

PONE-D-19-19157R1 

Characterization of Worldwide Olive Germplasm Banks of Marrakech (Morocco) and Córdoba (Spain): towards management and use of olive germplasm in breeding programs 

Dear Dr. Khadari:

I am pleased to inform you that your manuscript has been deemed suitable for publication in PLOS ONE. Congratulations! Your manuscript is now with our production department. 

With kind regards,

on behalf of

Dr. Randall P. Niedz 

Academic Editor

PLOS ONE